# An improved GC-MS-SIM analytical method for determination of pendimethalin residue in commercial crops (leaf and soils) and its validation

Anindita Paul[1☯*], Sujan Majumder[2☯], L. K. Prasad[1], M. Sheshu Madhav[3], Nalli Johnson[1☯], K. Padmaja[1], Satyapriya Singh[4]

1 Division of Post harvest & Value Addition, ICAR- National Institute for Research on Commercial Agriculture, Rajahmundry, Andhra Pradesh, India, 2 ICAR-Indian Institute of Vegetable Research, Varanasi, Uttar Pradesh, India, 3 Director, ICAR- National Institute for Research on Commercial Agriculture, Rajahmundry, Andhra Pradesh, India, 4 Central Horticultural Experiment Station (ICAR-IIHR), Bhubaneswar, Odisha, India

☯ The authors contributed equally
* aninditapaul20@gmail.com

## Abstract

Pendimethalin is generally used as pre-emergence herbicide cum suckericide in different growing region of Flue-Cured Verginia (FCV) tobacco by applying indiscriminately in irrigation channel. In quest of safety regulations related to pendimethalin residues in cured leaf tobacco and tobacco growing soils, a rapid and robust method is developed by GC-MS single quadruple system.. The GC-MS single ion monitoring (SIM) analytical method, achieved good linearity ($R^2 > 0.99$) with LOD and LOQ values of 0.001 mg/kg and 0.005 mg/kg, respectively. The method gave more than 80% recovery (5% RSD) showed compliance with international specification of DG-SANTE guidelines. In the majority of samples, pesticide residue levels were below the guidance residue levels (GRL; for pendimethalin GRL 5 ppm) value set by Center for Scientific Research Relative to Tobacco (CORESTA). The analytical method developed for true detection of pendimethalin residue at very trace levels with acceptable recovery level and matrix effect. The method is improved in terms of sentivity and precision as per regulatory norms of exporting commercial crops like tobacco. Looking at the consumer safety the method can be used in monitoring the pendimethalin residues in FCV tobacco and FCV tobacco grown soils at regular intervals.

## Introduction

Tobacco (*Nicotiana tabacum* L.), one of the world's leading commercial crops, is grown extensively in countries such as India, China, Brazil, USA, etc. Tobacco is being cultivated in an area of about 0.47 million hectares, accounting for 0.33 per

**Data availability statement:** All data are in the paper and/or Supporting information files.

**Funding:** The authors are also thankful to the Council of Scientific & Industrial Research (CSIR) for assistance and for supporting the research work.

**Competing interests:** NO authors have competing interests.

cent of the total arable land in the country (http://www.fao.org/faostat/en/#compare). The increasing demand for the Indian tobacco made the crop to be the major revenue source to India with an export value of US$ 859.45 million. India is maintaining its dominant position in the world as third largest producer of tobacco (~800 million kg) after China and Brazil (https://tobaccoboard.com/indexeng.php). In countries like India, with tropical-humid climate, the incidences of pest and disease infestations are frequent and application of pesticides for their management is almost obligatory. The pesticides applied to tobacco during its cultivation may remain in the leaf till harvesting and even after curing/ post-harvest processing in the final manufactured product [1–3]. While smoking, the pyrolytic products from tobacco plant matrix may interact with the pesticides or with the pyrolytic products of pesticides and results in the formation of more toxic smoke than that from the sole pesticide residues which is being inhaled by mainstream smoker and several other passive smokers [4]. Clapp and Shelar (1972) reported that rate of transfer of pesticides from tobacco into smoke averaged about 12% of that in the tobacco before combustion [5]. Transfer of suckericide like pendimethalin residue to plant leaf system may reach up to 20% of the residues in the unsmoked tobacco as this pesticide is mainly applying indiscriminately in irrigation channel at 90 days after planting which get translocated to soil and plant system very rapidly [6–8]. Looking into the both the apprehensions related to accumulation of pesticide residues at toxic levels in the final produce drives the safety regulations to become more and more stringent in most countries. Despite several awareness campaigns about the imminent potential health problems associated with tobacco, millions of people, particularly in lower and middle-income societies still indulge in cigarette smoking in this global COVID 19 pandemic [9]. Presence of pesticide residues in tobacco further aggravates the health risk not only to the smoker, but also to those subject to passive inhalations. Thus, monitoring of pesticide residues in tobacco is an important issue of critical concern from public health and safety point of view demanding implementation of stringent regulatory policies [10].

The Guidance Residue Levels (GRL) set by Cooperation Center for Scientific Research Relative to Tobacco [10], list for tobacco contains different classes of pesticides, such as organochlorine, organophosphorus, pyrethroids etc. Pendimethalin, a pesticide of the dinitroaniline class, is widely used as an herbicide in tobacco cultivation across the world, including India. However, its use as a suckericide for sucker control in tobacco has drawn attention of the scientific community as well as of the exporting agencies. Hence, determination of pendimethalin residue is an urgent need as there is no report on Indian tobacco.

The complex nature of tobacco matrix, led many obstacles during sample preparations and method accuracy very less while doing analysis by GC [11–13]. Very few literature is available in pertaining pendimethalin residue analysis in tobacco matrix with selective determination by GC MS [14]. The previously developed method owned time consuming acquisition and unnecessarily incurred the costing of method analysis and more over the quantification limit is higher than 0.01 mg/kg which demands the improvisation of the existing method in compliances with GRL (Guidance Residue level) to target the trade barrier associated with tobacco.

The objective of the present study was to develop a sensitive, effective, and economic analytical method for pendimethalin pesticide in tobacco using a GC-MS single quadrupole instrument. In addition to the optimization of the selected ion monitoring (SIM) parameters, the technique of high-resolution GC-MS is demonstrated in resolving complex matrix effect problems.

## Experimental

### Collection of tobacco samples

The cured leaf samples of Flue Cured Virginia (FCV) tobacco were collected from Andhra Pradesh and Karnataka, the two major tobacco producing states of India. In all 60 representative leaf and soil samples were drawn from major FCV tobacco growing districts, i.e., *Nellore* (14.44° N 79.99° E), *Prakasam* (15.50° N 80.05° E), *West Godavari* (16.57° N 82.15° E) of Andhra Pradesh and Mysore (12.08° N 76.32° E) district of Karnataka. After removing the midribs of the representative leaf samples, around 50 g of leaf lamina were oven dried at 60° C for 2 h. The dried leaves were powdered, homogenized, passed through 1-mm sieve, and utilized for further analyses. The soil samples were air-dried, homogenised and passed through 2 mm sieve for further analyses. The ICAR-National Institute for Research on Commercial Agriculture has given the permission to work and collection of the sample from the particular zones.

### Chemicals and apparatus

The certified reference standard of pendimethalin (98.9% purity) was procured from M/S Sigma-Aldrich Pvt. Ltd., India. Ethyl acetate of analytical grade – the extraction solvent, anhydrous sodium sulphate ($Na_2SO_4$), magnesium sulphate ($MgSO_4$) and glacial acetic acid (Analytical reagent grade) were procured from Thomas Baker (Mumbai, India). GC-MS grade ethyl acetate was procured from Merck (Mumbai, India). The adsorbents, Primary secondary amine (PSA), graphitized carbon black (GCB) and $C_{18}$ were procured from Agilent Technologies, Bangalore, India. Precision balance (Vibra, Adair Dutt, Mumbai, India), vortex mixer (Spinix, Borosil, Mumbai, India), micro centrifuge (Borosil R Mumbai, India), centrifuge (Kubota, Germany) and ultra sonicator bath (Oscar electronics, Mumbai, India) were used while preparation of samples and reagents.

### Preparation of standard solutions and matrix matched standards

The standard stock solutions of pendimethalin were prepared by weighing 10 (± 0.01) mg of the reference standards dissolving in 10 mL ethyl acetate in a certified 'A' class volumetric flask. The resulted final concentration of ~ 1000 µg/mL stored in dark in a refrigerator at −20 (±2) °C. An intermediate working standard mixture of 10 µg/mL was prepared by mixing appropriate quantities of the standard stock in ethyl acetate constancy of the working standard solution was checked against freshly prepared working standards (1 µg/mL) from the intermediate standards as per SANTE/11312/2021 guidelines [15]. From the 1 µg/mL working standard solution different calibration standards (0.001–0.08 µg/mL) were prepared by serial dilution with ethyl acetate. The matrix matched standards at the same concentration were prepared by extracting control tobacco matrix and soils and spiking the extract with appropriate volumes of the working standard solutions.

### Procedure of Extraction and clean-up

The entire sample were mixed thoroughly and selected randomly for further analysis for both tobacco leaf powder and soil. The samples were extracted by following earlier reported method with minor modifications [16]. Samples of 20 g homogenate (2 g tobacco + 18 mL water containing 0.5% acetic acid) were extracted by 10 mL ethyl acetate (vortex for 1 min) followed by the addition of 10 g $Na_2SO_4$ (vortex for 1 min). This was subjected to phase separation by centrifugation at 5000 rpm for 5 min. A supernatant of 2 mL ethyl acetate was cleaned up using two different combinations of dispersive solid-phase extraction (*d*SPE) sorbents as follows: 25, 50, 75 and 100 mg of PSA per mL of supernatant. Thereafter the

combinations of $C_{18}$ and GCB with $MgSO_4$, were evaluated to get the good and acceptable results. The supernatant of the above extract (1 mL) was centrifuged at 10000 rpm for 5 min. It was further filtered through a 0.2 µm polytetrafluoroethylene (PTFE) membrane filter and finally injected (1 µL) into the GC-MS system and analysed in SCAN mode with reference standards (based on retention time) and in SIM mode with quantifier and qualifier ions (*m/z*). Soil samples were also extracted using the same aforementioned sample preparation workflow.

## Instrumentation

The GC-MS QP-2010 Plus (single quadrupole, Shimadzu Corporation, Kyoto, Japan) system was equipped with ZB-5 (5% diphenyl, 95% dimethylpolysiloxane, 30 m (l) x 0.25 mm (id), 0.1 µm film thickness) capillary column and autosampler. The GC-MS separation of Pendimethalin was achieved by an optimized oven temperature program that started from an initial temperature of 90 °C (hold for 0.5 min), ramped at the rate of (@) 20 °C min$^{-1}$ up to 180 °C (hold 1 min), increased to 240 °C @ 12 °C min$^{-1}$ (hold for 1 min), again increased to 260°C @ 15 °C min$^{-1}$ (hold for 1 min) and finally increased up to 280°C @ 12 °C min$^{-1}$ (hold 0.5 min). This program resulted in total run-time of 16.5 min. The sample solutions were injected in split injection mode (split ratio 10 and pressure 29.1 psi for 1 min) with the injection volume of 1 µL. The injector temperature was set at 250°C. The ion source temperature was 200°C and the interface was at 280°C. The detector voltage was set at 0.87 kV and the data acquisition was carried out in the selected ion monitoring (SIM) mode with specific *m/z* ions for selective identification of pendimethalin. Ultra-pure (99.999%) grade helium (INOX Limited, Hyderabad) was used as the carrier gas. The flow rate of Helium was maintained as 3.14 mL/min with a linear velocity of 65.6 cm/sec. The mass spectrometer was operated using electron impact ionization (EI, 70 eV).

## Method performance

A single laboratory based method validation was performed as per SANTE/11312/2021 guideline which includes linearity, limit of detection (LOD), limit of quantification (LOQ), matrix effect, accuracy and repeatability and recovery. The calibration curves for linearity establishment for pendimethalin in pure solvent and in matrix (tobacco leaf and soil matrix) by establishing seven levels ranging between 0.001–0.08 mg/L (Fig 1 and S1 Fig 1 a,b,c).

 The sensitivity of the method was determined in terms of limit of detection (LOD) and limit of quantification (LOQ) which decides as the smallest measured quantity in tobacco leaf and soil matrix at which the signal to noise ratio (S/N) were 3:1 and 10:1, respectively. The method LOQ was the lowest concentration at which method was validate with acceptable

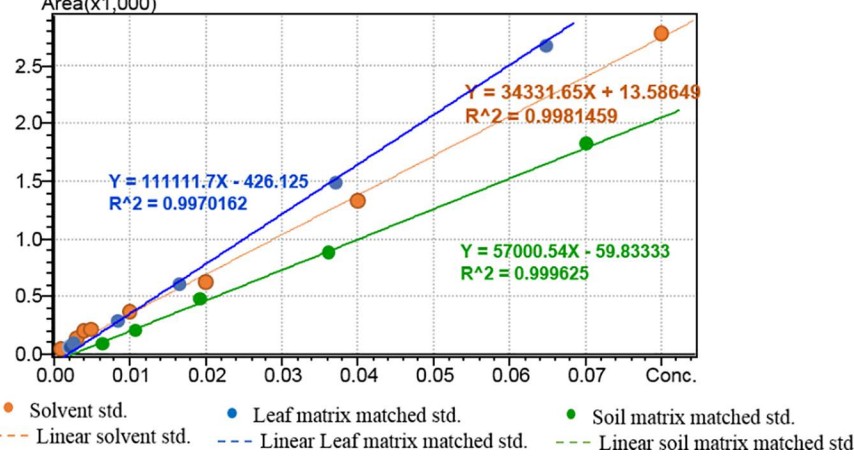

**Fig 1. Linearity calibration curve for solvent, tobacco leaf matrix matched and soil matrix matched standard.**

recoveries. Recovery of the pendimethalin from tobacco leaf and soil matrix was studied at 0.01, 0.02, 0.05, 0.10 and 0.50 mg/kg with six replicates each (Table 3). The average recoveries within the range of 80–120% are accepted for both the matrices (tobacco leaf and soil) as per SANTE/11312/2021 guideline [15]. The matrix effect was determined by spiking untreated extract with the targeted pesticide at 0.05 mg/kg level. The peak area response of pendimethalin in ethyl acetate solvent was compared with that of the corresponding peak area in the matrix matched standard at the same concentration.

*Matrix effect* (*ME%*) = [(*peak area of the matrix standard – peak area of the solvent standard*)/*peak area of the solv*

A positive or negative value of ME (%) indicates that matrix induced signal enhancement and suppressions respectively [17–18].

## Analysis of incurred samples

We have collected FCV tobacco samples from farmers' field of Nellore, Prakasam and West Godavari district of Andhra Pradesh and estimated pendemethalin residues using above mentioned standardized extraction and analytical method.

## Results and discussion

### Optimization of extraction and clean up method

Residue study in tobacco leaf and soil is highly essential for promoting residue management and ensure safe and rational use of pesticide in tobacco. It will also help for regulatory analysis of FCV tobacco which will promote the Indian tobacco exports in compliance with the acceptable levels of residues. Extraction of targeted compounds from the tobacco matrix is very challenging task [16]. The ethyle ecetate were selected for the extraction of the targeted compounds, which provided accepted results in the previously reported studies [19,20]. However, acetic acid is generally used to stabilize several natural compounds present in tobacco matrix during sample preparation which interferes in the response of pendimethalin as a false positive [16,19]. I, the extract was injected without cleanup and as a results the yellow color pigments of tobacco leaf extract were highly reducing the recovery percentage of pendimethalin of about 2.5–3%. On the other hand, it was causing interference in GC-liner which forced to change the liner frequently. Therefore, cleanup plays a key role in sample preparation in case of complex tobacco matrix. To reduce the matrix induced signal enhancement, the ethyl acetate extract of tobacco leaf and soil matrix was undergone to clean up process with PSA, which provided lower matrix effect up to some extent (12.34%) with 100 mg of PSA+200 mg MgSO$_4$. However to further reduce the matrix effect and matrix interference some combination were tried with 100 mg PSA + (25, 50, 75 and 100 mg) C$_{18}$ + (5, 10, 25, 50 and 100 mg) GCB alone with 200 mg of MgSO$_4$ (S1 Table). which can reduce the matrix effect to less than17%. Though initially, with 100 mg GCB gave charcoal interference during injection whereas lesser than 50 mg GCB gave more colored solution resulting ghost peaks along with target compounds with matrix effect upto 46%. Finally, the combination of 100 mg PSA, 100 mg C$_{18}$, 50 mg GCB and 200 mg MgSO$_4$ provided the lowest matrix effect for the targeted compound and also which helped to reduce frequent cleaning as well as change of GC-Liner.

Hence cleanup of 2 mL ethyl acetate extract of tobacco leaf and soil matrix were optimized with 100 mg PSA, 100 mg C$_{18}$, 50 mg GCB and 200 mg MgSO$_4$. The one of the novelty of this study is as tobacco is a complecated matrix and it is very difficult to have lesser matrix effect of targeted compound, since the extraction method is need to be optimized along with analytical parameters which is discussed below.

This method provided accepted recovery for tobacco leaf ranging from 87–95.34% with 2.73–8.15% RSD and in case of soil 80.01–84.99% with 1.42–6.25% RSD which complies to the analytical quality control guideline [10,15].

### Comparative performance with previously developed method

Earlier approaches of pendimethalin estimation in complecated matrix like peanut, tobacco involves greater amount of clean up reagents which not only incurred more cost but also induces indirect matrix intereferences (10–34%) [19,22,23]. More over the GC-MS based methods also not so robust such as the acqusition parameters are time consuming, the transitions were not well defined. More over, the LOQ levels are higher upto 0.01 mg/kg which in turns suggest the

redevelopment of pendimethalin estimation in tobacco like complicated matrix. On the other hand, pendimethalin residues in the incurred samples were not estimated earlier [24]. The present study shaded light into these drawbacks and re developed a new method for estimating pendimethalin residues in both leaf and soil to interrogate the residues present in incurred samples and as well as the acquisition method is faster and robust having LOQ level of 0.005 mg/kg for both leaf and soil matrices. The developed method itself given an insight of fate of utilization of higher GCB and C18 and it was further optimized to obtain better recovery% and lower matrix effect (Fig 2). In this way, the developed method stands out than the existing method to estimate pendimethalin in complex matrices.

## Confirmative analysis by GC-MS

The probabilities of acquiring false detection in plant or soil matrices become very crucial as they may act as interfering components in deferring the response of target pesticide at the retention time while using GC-MS instrument. Hence to deescalate the false detection, a novel GC-MS based selective ion monitoring (SIM) method has developed employing with different acquisition parameters such as confirmative identification based on the quantifier-qualifier ions (m/z) ratio. Depending on the molecular fragmentation recorded in the mass detection system, four ions (*m/z*) namely, 162, 191, 252 and 281 were selected. However, the reference and target ion combination resulted a complex chromatogram where, the base ion, 162, can exist in several other molecules. Hence, the ion 162 was ejected as base ions and m/z 252 was selected as quantifier ion and rest four ions (162, 191, 208, 252, and 281) were selected as qualifier ions (Fig 3a, 3b and 4) for selective identification of pendimethalin.

The results of Table 1 and 2 indicated that the pendimethalin residues detected in both tobacco leaf and soil are in the range of 0.0052–0.0169 mg kg$^{-1}$ which are far below than GRL level (5 mg kg$^{-1}$) of pendimethalin in tobacco (8). All the samples achieved precision >91% with good repeatability.

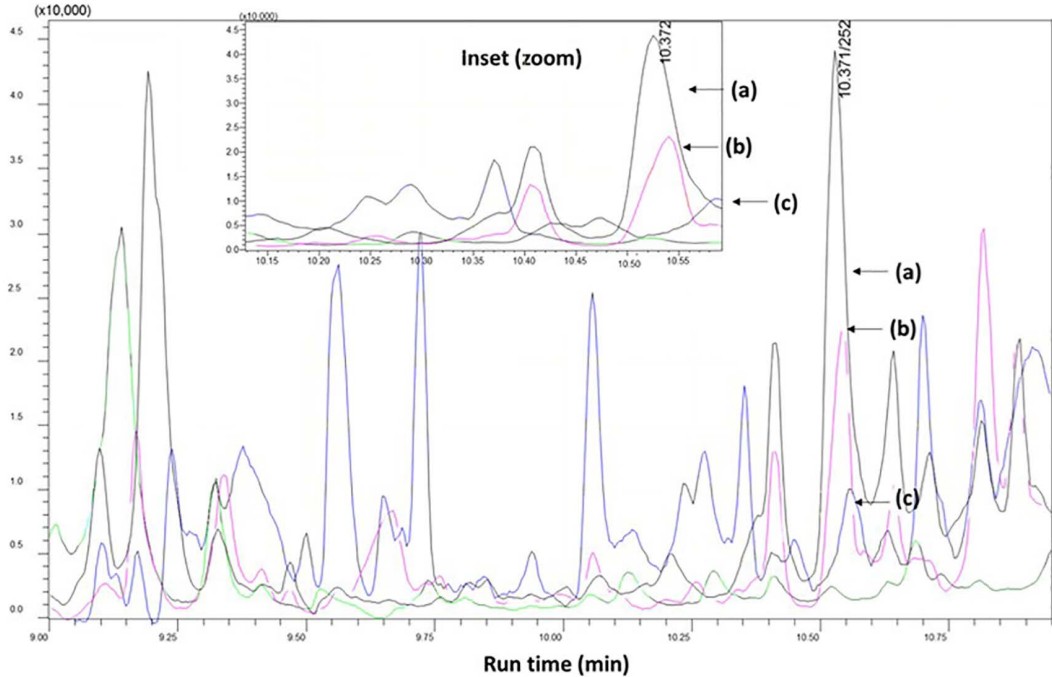

**Fig 2. Matrix effect during determination of pendimethalin.** (a) pendimethalin response in solvent standard at 0.01 mg/kg (b) pendimethalin response in tobacco leaf matrix matched standard at 0.01 mg/kg (c) pendimethalin response in soil matrix matched standard at 0.01 mg/kg.

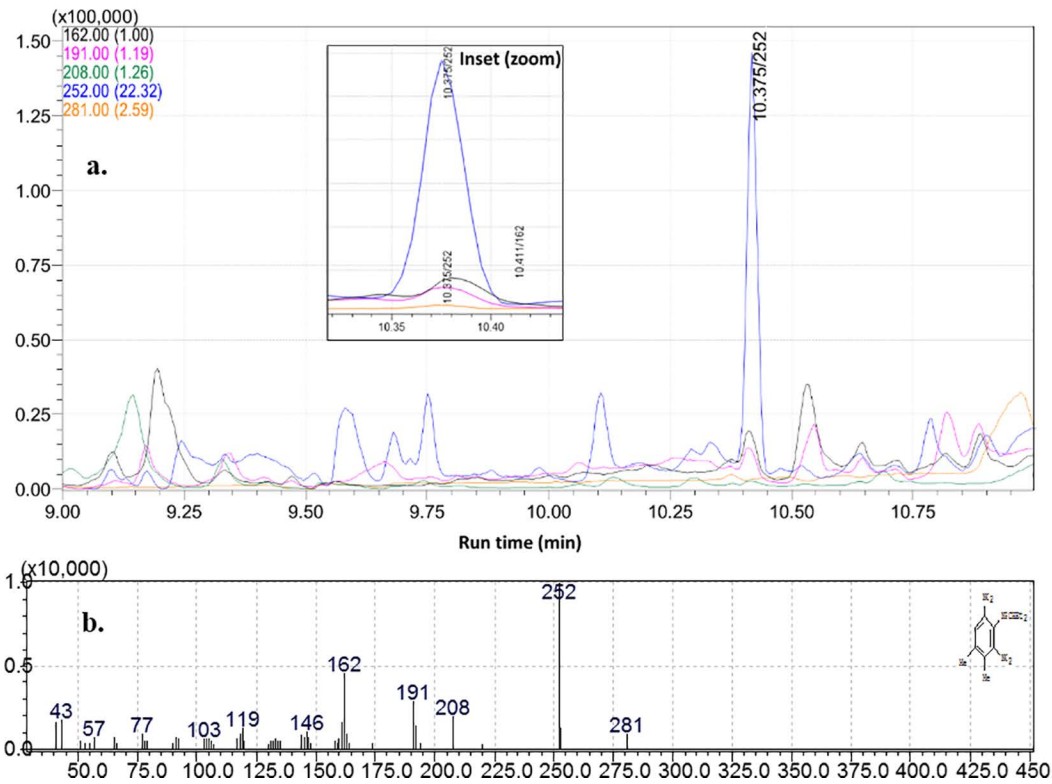

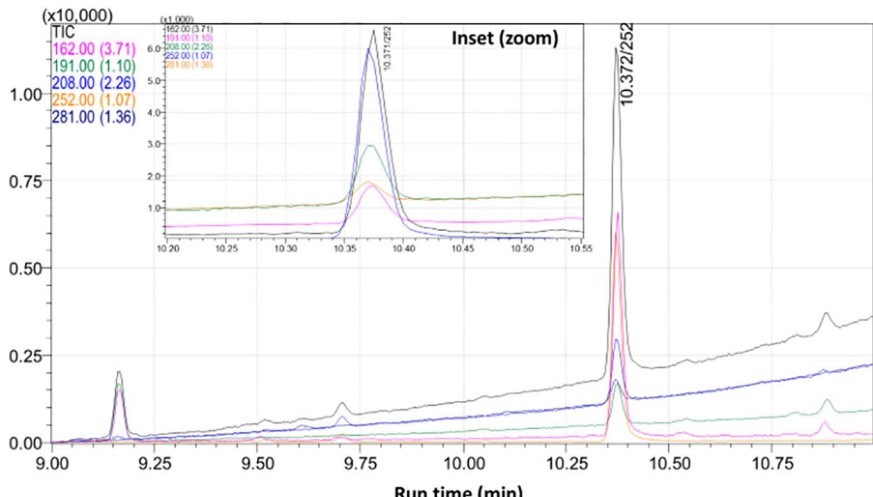

**Fig 3. Different ion transition for quantification of pendimethalin in tobacco leaf matrix.** (a) 162, 191,208, 252& 281 ions (b) mass fragmentation pattern in Selection Ion Monitoring (SIM) mode.

**Fig 4. Different ion transition for quantification of pendimethalin in soil matrix.** It was observed that selection of 252 ion as a base peak improved the identification and quantification of pendimethalin as compared to 162 ion. The developed method was employed for analysis of some field samples (Fig 5).

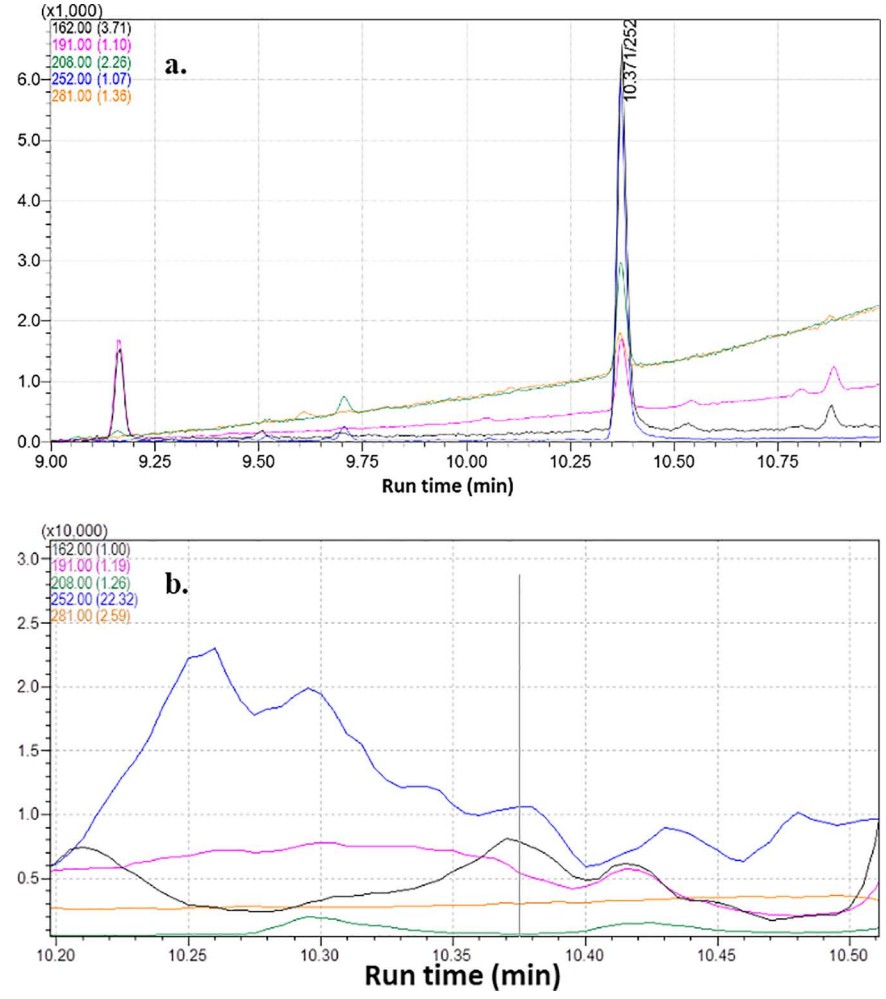

**Fig 5. Sample analysis for estimation of pendimethalin with the newly developed method.** (a) sample with maximum pendimethalin residue present and (b) sample with BDL level of pendimethalin residue present.

## Method validation

**LOD, LOQ and recovery.** The analytical method for estimation of pendimethalin residues in tobacco leaf and soil was validated according to SANTE guideline. Pendimethalin had a retention time (Rt) of 10.37 min in newly developed analytical method. The good linearity achieved with coefficient of determination ($R^2$) value of 0.998 for solvent standard, 0.997 for tobacco leaf matrix matched standard and 0.999 for soil matrix matched standard within the calibration range of 0.001–0.08 mg/kg. The LOQ for pendimethalin were established as and 0.005 mg/kg for both the matrices, which were far below the GRL value (5 mg/kg). The percentage recovery was estimated at five levels for both tobacco leaf as well as soil matrices separately. The percentage recoveries at 0.01, 0.02, 0.05, 0.10 and 0.50 mg/kg were found in the range between 80–95% for both the matrices (Table 3) with relative standard deviation (% RSD) less than 20% which met internationally accepted SANTE guidelines. The average matrix effect (ME) percentage were less than ~10% for leaf and ~11% for soil matrix. Off late, Paul et al, 2021 had established a precise liquid chromatographic with high resolution mass spectrometry (Orbitrap) method for detection of multiple pesticide residues in tobacco matrix and achieved good screening detection limit at 5 ng/g level which fulfilled the requirements of analytical quality control guideline of SANTE/11312./2021 [15]. Off late, Rahman et al. (2012)

**Table 1. Pendimethalin residues present in FCV tobacco leaf samples analyzed by GC-MS/SIM method.**

| Sample Details | Location | Pendimethalin concentration (mg kg$^{-1}$) |
|---|---|---|
| SLS Tobacco 22 | Podili, (15.6070° N, 79.6146° E) Prakasam district of Andhra Pradesh | 0.00817 |
| SLS Tobacco 23 | Podili, (15.6070° N, 79.6146° E) Prakasam district of Andhra Pradesh | 0.00912 |
| SLS Tobacco 25 | Kandukur, (15.2197° N, 79.9025° E) Prakasam district of Andhra Pradesh | 0.00324 |
| SLS Tobacco 26 | Kandukur, (15.2197° N, 79.9025° E) Prakasam district of Andhra Pradesh | 0.00771 |
| SLS Tobacco 27 | Kandukur, (15.2199° N, 79.9027° E) Prakasam district of Andhra Pradesh | 0.00524 |
| SLS Tobacco 28 | Kaligiri, (14.8279° N, 79.6937° E) Prakasam district of Andhra Pradesh | 0.00976 |
| SLS Tobacco 29 | D C Palli, (14.6876° N, 79.5034° E) Prakasam district of Andhra Pradesh | BDL |
| SLS Tobacco 30 | D C Palli, (14.6876° N, 79.5034° E) Prakasam district of Andhra Pradesh | BDL |
| SLS Tobacco 32 | Kanigiri, (15.4048° N, 79.5067° E) Prakasam district of Andhra Pradesh | BDL |
| SLS Tobacco 33 | Kanigiri, (15.4048° N, 79.5067° E) Prakasam district of Andhra Pradesh | BDL |
| SLS Tobacco 35 | Kanigiri, (15.4048° N, 79.5067° E) Prakasam district of Andhra Pradesh | 0.00987 |
| NLS Tobacco 32 | Jangareddygudem, (17.1223° N, 81.2923° E) West Godavari district of Andhra Pradesh | 0.00859 |
| NLS Tobacco 17 | Devarapalli, (17.0350° N, 81.5624° E) West Godavari district of Andhra Pradesh. | 0.01691 |
| NLS Tobacco 30 | Koyyalagudem, (17.4521° N, 81.6528° E) West Godavari district of Andhra Pradesh | BDL |
| NLS Tobacco 32 | Jangareddygudem, (17.1223° N, 81.2923° E) West Godavari district of Andhra Pradesh | BDL |
| NLS Tobacco 33 | Jangareddygudem, (17.1223° N, 81.2923° E) West Godavari district of Andhra Pradesh | 0.00505 |
| NLS Tobacco 34 | Jangareddygudem, (17.1223° N, 81.2923° E) West Godavari district of Andhra Pradesh | 0.00505 |
| NLS Tobacco 36 | Koyyalagudem, (17.4521° N, 81.6528° E) West Godavari district of Andhra Pradesh | 0.00305 |
| NLS Tobacco 37 | Jangareddygudem, (17.1223° N, 81.2923° E) West Godavari district of Andhra Pradesh | BDL |
| NLS Tobacco 38 | Jangareddygudem, (17.1223° N, 81.2923° E) West Godavari district of Andhra Pradesh | BDL |
| NLS Tobacco 40 | Koyyalagudem, (17.4521° N, 81.6528° E) West Godavari district of Andhra Pradesh | BDL |
| NLS Tobacco 42 | Koyyalagudem, (17.4521° N, 81.6528° E) West Godavari district of Andhra Pradesh | BDL |
| KLS Tobacco 5 | Periyapatna, (12.3384° N, 76.0965° E) Mysore district of Karnataka district of Karnataka. | 0.00501 |
| KLS Tobacco 4 | Periyapatna, (12.3384° N, 76.0965° E) Mysore district of Karnataka district of Karnataka. | 0.00842 |
| KLS Tobacco 7 | Ramnathapura, (12.6166° N, 76.0842° E) Mysore district of Karnataka. | BDL |
| KLS Tobacco 3 | Hunsur, (12.3091° N, 76.2833° E) district Mysore of Karnataka | BDL |
| KLS Tobacco 63 | Ramnathapura, (12.6166° N, 76.0842° E) Mysore district of Karnataka. | 0.00406 |
| KLS Tobacco 64 | Ramnathapura, (12.6166° N, 76.0842° E) Mysore district of Karnataka. | 0.00506 |
| KLS Tobacco 65 | Ramnathapura, (12.6166° N, 76.0842° E) Mysore district of Karnataka. | BDL |
| KLS Tobacco 67 | Ramnathapura, (12.6166° N, 76.0842° E) Mysore district of Karnataka. | BDL |
| KLS Tobacco 70 | Ramnathapura, (12.6166° N, 76.0842° E) Mysore district of Karnataka. | BDL |
| KLS Tobacco 73 | Ramnathapura, (12.6166° N, 76.0842° E) Mysore district of Karnataka. | 0.00389 |
| KLS Tobacco 80 | Ramnathapura, (12.6166° N, 76.0842° E) Mysore district of Karnataka. | 0.00132 |
| KLS Tobacco 82 | Ramnathapura, (12.6166° N, 76.0842° E) Mysore district of Karnataka. | 0.00897 |
| KLS Tobacco 84 | Ramnathapura, (12.6166° N, 76.0842° E) Mysore district of Karnataka. | 0.00629 |
| KLS Tobacco 86 | Ramnathapura, (12.6166° N, 76.0842° E) Mysore district of Karnataka. | 0.00498 |

*BDL=Below Detection Limit; NLS: Northern light soils; SLS; Southern Light Soils; KLS; Karnataka light soils.

and Chen et al., 2013 developed a high performance liquid chromatography (HPLC) method in detecting some pesticides with only single acquisition parameter. Particularly, many plant matrix compound can elute at same retention time with that of same target pesticide called false detection and the concentration of plant matrix interfering compound is more prevalent than the target pesticide [19]. To resolve the problem mass fragmentation can give the confirmative analysis to eject the falsified detection induced by matrices [20–21].

**Table 2. Pendimethalin residues present in soil samples analysed by GC-MS/SIM method.**

| Sample Details | Location | Pendimethalin concentration (mg kg⁻¹) | |
|---|---|---|---|
| BTJ S2 | Podili, (15.6070° N, 79.6146° E) Prakasam district of Andhra Pradesh | BDL | |
| BTJ S3 | Kandukur, (15.2199° N, 79.9027° E) Prakasam district of Andhra Pradesh | BDL | |
| BTJ S4 | Jangareddygudem, (17.1223° N, 81.2923° E) West Godavari district of Andhra Pradesh | BDL | |
| BTJ S7 | Kaligiri, (14.8279° N, 79.6937° E) Prakasam district of Andhra Pradesh | BDL | |
| BTJ S9 | Koyyalagudem, (17.4521° N, 81.6528° E) West Godavari district of Andhra Pradesh | BDL | |
| BTJ S10 | Koyyalagudem, (17.4521° N, 81.6528° E) West Godavari district of Andhra Pradesh | BDL | |
| BTJ S16 | Koyyalagudem, (17.4521° N, 81.6528° E) West Godavari district of Andhra Pradesh | BDL | |
| BTJ S21 | Koyyalagudem, (17.4521° N, 81.6528° E) West Godavari district of Andhra Pradesh | BDL | |
| BTJ S25 | Kaligiri, (14.8279° N, 79.6937° E) Prakasam district of Andhra Pradesh | BDL | |
| BTJ S27 | Kaligiri, (14.8279° N, 79.6937° E) Prakasam district of Andhra Pradesh | BDL | |
| BTJ S28 | Kaligiri, (14.8279° N, 79.6937° E) Prakasam district of Andhra Pradesh | BDL | BTJ S25 |
| BTJ S40 | Kaligiri, (14.8279° N, 79.6937° E) Prakasam district of Andhra Pradesh | BDL | BTJ S25 |
| BTJ S45 | Kaligiri, (14.8279° N, 79.6937° E) Prakasam district of Andhra Pradesh | 0.0002 | BTJ S25 |
| BTJ S48 | Kaligiri, (14.8279° N, 79.6937° E) Prakasam district of Andhra Pradesh | BDL | BTJ S25 |
| BTJ S50 | Kaligiri, (14.8279° N, 79.6937° E) Prakasam district of Andhra Pradesh | 0.0004 | BTJ S25 |
| BTJ S29 | Jangareddygudem, (17.1223° N, 81.2923° E) West Godavari district of Andhra Pradesh | 0.0018 | |
| BTJ S30 | Jangareddygudem, (17.1223° N, 81.2923° E) West Godavari district of Andhra Pradesh | 0.00002 | |
| BTJ S49 | Jangareddygudem, (17.1223° N, 81.2923° E) West Godavari district of Andhra Pradesh | BDL | |
| BTJ S53 | Jangareddygudem, (17.1223° N, 81.2923° E) West Godavari district of Andhra Pradesh | BDL | |
| BTJ S54 | Jangareddygudem, (17.1223° N, 81.2923° E) West Godavari district of Andhra Pradesh | BDL | |
| BTJ S89 | Podili, (15.6070° N, 79.6146° E) Prakasam district of Andhra Pradesh | 0.0008 | |
| BTJ S90 | Podili, (15.6070° N, 79.6146° E) Prakasam district of Andhra Pradesh | 0.0001 | |
| BTJ S94 | Podili, (15.6070° N, 79.6146° E) Prakasam district of Andhra Pradesh | BDL | |
| BTJ S97 | Podili, (15.6070° N, 79.6146° E) Prakasam district of Andhra Pradesh | BDL | |

*BDL=Below Detection Limit.

**Table 3. Recovery percentage of pendimethalin in tobacco leaf and soil.**

| Level of fortification (mg kg⁻¹) | % Recovery | % RSD |
|---|---|---|
| Tobacco leaf | | |
| 0.01 | 87.00 | 3.15 |
| 0.02 | 89.33 | 6.93 |
| 0.05 | 90.67 | 2.73 |
| 0.1 | 95.34 | 8.15 |
| 0.5 | 90.89 | 3.67 |
| Soil | | |
| 0.01 | 83.54 | 1.43 |
| 0.02 | 80.01 | 6.25 |
| 0.05 | 83.09 | 2.38 |
| 0.1 | 84.99 | 5.73 |
| 0.5 | 83.01 | 0.00 |

**Analysis of incurred samples.** The analysed samples found to have no residues of pendimethalin as residues estimated well below the GRL (5 mg/kg) which clearly indicated that the farmers of those representative districts of Andhra Pradesh are following good agricultural practices.

## Conclusions

The novel analytical method developed for true detection of pendimethalin residue at very trace levels which satisfy the internationally accepted guidelines with acceptable recovery level and matrix effect. The new method achieved good linearity ($R^2 > 0.99$) over the concentration range of 0.001–0.08 mg kg$^{-1}$ had limit of determination (LOD) and limit of quantification (LOQ) values of 0.001 mg kg$^{-1}$ and 0.005 mg kg$^{-1}$, respectively. The method with more than 80% recovery (with 5% RSD). The method ensures sensitive and accuriate detection of pendimehalin residues in tobacco leaf and soil samples. The studies showed that the majority of samples, contain pendimethalin pesticide residue levels far below the GRL value set by CORESTA. The ethyl acetated based extraction method resulted good recovery with good repeatability. It also shows that the levels of pendimethalin residues in FCV tobacco are very low to negligible level and pendimethalin used across FCV tobacco as suckericide does not result residue above critical level. It also holds promise in facilitating selective and sensitive residue analyses of pesticides in such a complicated matrix like tobacco and resolving false detection. This method is suitable and will be applicable for routine analysis of pesticide residues in tobacco samples as well as soil samples. Looking at the consumer safety the method can be used in monitoring the pendimethalin residues in FCV tobacco and FCV tobacco grown soils at regular intervals.

## Supporting information

**S1 Table. Comparative matrix effect of different combination of clean up dring pendimethalin estimation in tobacco leaf.**
(DOCX)

**S1 Fig. Linearity calibration curve.** (a) for solvent standard (b) for tobacco leaf matrix matched standard (c) soil matrix matched standard.
(DOCX)

## Acknowledgments

The authors are thankful to Director, The ICAR-National Institute for Research on Commercial Agriculture, Rajahmundry for support to carry out this research.

## Author contributions

**Conceptualization:** Anindita Paul.

**Data curation:** K. Padmaja.

**Formal analysis:** Sujan Majumder, Nalli Johnson.

**Investigation:** Satyapriya Singh.

**Project administration:** L. K. Prasad.

**Supervision:** M. Sheshu Madhav.

**Visualization:** Anindita Paul.

**Writing – original draft:** Anindita Paul.

**Writing – review & editing:** Anindita Paul, Sujan Majumder, M. Sheshu Madhav.

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
