## [Decision Letter · Decision Letter 0]

7 Apr 2025

Dear Dr. Paul,

Thank you for submitting your manuscript to PLOS ONE. After careful consideration, we feel that it has merit but does not fully meet PLOS ONE’s publication criteria as it currently stands. Therefore, we invite you to submit a revised version of the manuscript that addresses the points raised during the review process.

We look forward to receiving your revised manuscript.

Kind regards,

Vinaya Satyawan Tari, Post doctoral fellow, (M.Sc., B.Ed., Ph.D.)

Academic Editor

PLOS ONE

Additional Editor Comments (if provided):

Reviewers' comments:

Reviewer's Responses to Questions

**Comments to the Author**

1. Is the manuscript technically sound, and do the data support the conclusions?

Reviewer #1: Yes

Reviewer #2: Yes

2. Has the statistical analysis been performed appropriately and rigorously?

Reviewer #1: Yes

Reviewer #2: Yes

3. Have the authors made all data underlying the findings in their manuscript fully available?

Reviewer #1: Yes

Reviewer #2: Yes

4. Is the manuscript presented in an intelligible fashion and written in standard English?

Reviewer #1: Yes

Reviewer #2: Yes

Reviewer #1: This study assessed an improved GC-MS-SIM analytical method for determination of pendimethalin residue in commercial crops (leaf and soils) and its validationwhich his very important from high environment and safety requirements point of views.

However minor corrections are required (already mentioned in manuscript pdf file).

Reviewer #2: 1. Considering that the QUCHERS method uses acetonitrile, why did the authors use ethyl acetate?

2. In Figure 1 Slope of calibration curve equation for soil matrix and solvent must be checked. (slope of green line must be lower than ted line).

3. Title must be changed to : An improved and validation GC-MS-SIM analytical method for determination of pendimethalin residue in commercial crops (leaf and soils)

4. In validation repeatability and reproducibility of method must be give.

**Do you want your identity to be public for this peer review?** For information about this choice, including consent withdrawal, please see our Privacy Policy

Reviewer #1: No

Reviewer #2: No

---

## [Author Response · Author response to Decision Letter 1]

17 Apr 2025

I am submitting the revised version of the manuscript by addressing the points raised during the review process.

---

## [Decision Letter · Decision Letter 1]

2 Jul 2025

An improved GC-MS-SIM analytical method for determination of pendimethalin residue in commercial crops (leaf and soils) and its validation

PONE-D-25-08615R1

Dear Dr. Paul,

We’re pleased to inform you that your manuscript has been judged scientifically suitable for publication and will be formally accepted for publication once it meets all outstanding technical requirements.

Kind regards,

Vinaya Satyawan Tari, Post doctoral fellow, (M.Sc., B.Ed., Ph.D.)

Academic Editor

PLOS ONE

Reviewers' comments:

Reviewer's Responses to Questions

**Comments to the Author**

Reviewer #1: All comments have been addressed

2. Is the manuscript technically sound, and do the data support the conclusions?

Reviewer #1: Yes

3. Has the statistical analysis been performed appropriately and rigorously?

Reviewer #1: I Don't Know

4. Have the authors made all data underlying the findings in their manuscript fully available?

Reviewer #1: Yes

5. Is the manuscript presented in an intelligible fashion and written in standard English?

Reviewer #1: Yes

Reviewer #1: (No Response)

**Do you want your identity to be public for this peer review?** For information about this choice, including consent withdrawal, please see our Privacy Policy

Reviewer #1: No

---

## [Editor Report · Acceptance letter]

PONE-D-25-08615R1

PLOS ONE

Dear Dr. Paul,

I'm pleased to inform you that your manuscript has been deemed suitable for publication in PLOS ONE. Congratulations! Your manuscript is now being handed over to our production team.

Kind regards,

on behalf of

Dr. Vinaya Satyawan Tari

Academic Editor

PLOS ONE